# Vibrational Analysis of Hydration-Layer Water around Ubiquitin, Unpeeled Layer by Layer: Molecular-Dynamics Perceptions

**DOI:** 10.3390/ijms232415949

**Published:** 2022-12-15

**Authors:** José Angel Martinez-Gonzalez, Prithwish K. Nandi, Niall J. English, Aoife Gowen

**Affiliations:** 1School of Chemical & Bioprocess Engineering, University College Dublin, Belfield, D04 V1W8 Dublin, Ireland; 2School of Biosystems Engineering, University College Dublin, Belfield, D04 N2E5 Dublin, Ireland; 3ISIS Pulsed Neutron and Moun Source, Rutherford Appleton Laboratory, Harwell Science & Innovation Campus, Chilton, Didcot OX11 0QL, UK; 4School of Pharmacy, Universidad San Pablo-CEU, CEU Universities, Urbanización Montepríncipe, 28660 Boadilla del Monte, Spain

**Keywords:** molecular dynamics, ubiquitin, interfacial water, power spectra, infrared spectra

## Abstract

Classical molecular-dynamics simulations have been performed to examine the interplay between ubiquitin and its hydration-water sub-layers, chiefly from a vibrational-mode and IR viewpoint—where we analyse individual sub-layers characteristics. The vibrational Density of States (VDOS) revealed that the first solvation sub-shell indicates a confined character therein. For layers of increasing distance from the surface, the adoption of greater bulk-like spectral behaviour was evident, suggesting that vibrational harmonisation to bulk occurs within 6–7 Å of the surface.

## 1. Introduction

Water proximal to material surfaces plays important and pivotal regulatory roles in different and disparate fields [1,2]. Indeed, the surrounding aqueous environment affects, often existentially, the folding, structure, mobility, and properties of a wide array of biomolecules, such as, amongst others, proteins, nucleic acids, and biological assemblies. Examination and scrutiny of these biosurface-proximal water molecules, called interfacial water, and the study of its ramifications has developed into an area of intense interest [3,4,5,6,7], and led to a broad range of open questions about how hydrophobic protein-surface regions bring about modulation in water organisation vis-à-vis its neat-liquid state—not least to how water molecules’ dynamical properties vary across the hydration layer abutting the protein.

Around solvated proteins, this interfacial water displays rather, and often spectacularly, different properties with respect to molecules in a bulk environment. These interfacial protein water molecules are not involved in crystallising or icing process per se; however, they are rendered quasi-immobile in the liquid state below the bulk-water melting point. These molecules in direct contact with the protein are in a sort of ‘glassy’ amorphous state, and adopt an important role since they induce broad and disparate biological activities in proteins [8], which otherwise are absent in anhydrous variants [9].

In recent decades, the dynamics of proteins and their surrounding hydration layers have been extensively studied, mainly focusing on elucidating that the protein-glass transition is related to the liquid-glass transition of water molecules in the related hydration layer [10,11]. Experimentally, the dynamic behaviour of the water abutting the interface has been detected by various spectroscopic techniques [12,13,14,15,16,17,18,19], like infrared spectroscopy and neutron methodologies. For example, neutron-scattering experiments for the evaluation of the vibrational density of states (VDOS) [12,20,21] and molecular-dynamics (MD) simulations sampling of the hydration molecules’ displacement [22] gives us information about interfacial-water mobility. 

However, experimental techniques cannot give rise to any real extent of meaningful information about the formation and breaking of hydrogen bonds—particularly in terms of underlying dynamics timescales. To understand the interaction between the atoms forming the outermost surface of the protein and the water molecules in contact therewith, and to quantify the shifting kinetics of these key stabilising hydrogen bonds, theoretical techniques are needed—mainly atomistic MD methods. In terms of still useful and more specialised experiments methods, THz-absorption spectroscopy has been proposed by Niabili and Havenith [23] to study the “dynamical hydration shell”, which includes all molecules that show different sub-picosecond water-network dynamics vis-à-vis such expected behaviour of their bulk-state counterparts. These Niabili-Havenith MD simulations show that the hydration water’s THz peak of an active site (80 cm^−1^) is blue-shifted relative to bulk water (by 30 cm^−1^), which is due to the interaction between water molecules that enhance the hydration shell [23,24]. In fact, more generally, combining Terahertz experiments and molecular-dynamics simulation are useful in the understanding of hydration-water behaviour [24,25] in an extended hydration shell (between 1 to 2 nm).

Recently, a combination of near-infrared spectroscopy with aquaphotomics has been proposed to explore the involvement of hydrogen bonds in the formation of layers enveloping proteins [26]. However, it is impossible to ascertain the actual solvation layer’s size using this method, and we need to leverage theoretical data to obtain the ratio of the different hydrogen bonds formed. In this sense, although the combination of spectroscopy and aquaphotomics is broadly promising, the lack of determination of solvation-layer size is a rather limiting and difficult problem, and exact hydrogen-bond interactions and kinetics cannot be isolated and scrutinised. In this respect, this points still to the need to apply MD techniques to dissect the details of these individual interactions at the fully atomistic scale. 

Our group has used MD recently to scrutinise the prototypical and well-studied egg white lysozyme (HEWL) protein’s translational self-diffusion and the hydrogen-bond dynamics underlying the fundamental behaviour of “hydration layers” (0–1.0 nm) [9,22,27,28]. It was found that there is a broad probability distribution of hydrogen-bond interactions, in terms of strength and lifetime, depending on water molecules’ location in the heterogeneous “patchwork” of surface interactions in HEWL—both hydro-phobic and -philic. In some cases, water molecules are effectively entrapped in surface ‘clefts’ and the strong hydrogen bonds thereat create de facto immobility, whilst in other cases, there is a good level of surface mobility, and exchange and swapping between outer parts of the broader hydration layer, i.e., dynamical transfer between its (sub-)shells. Indeed, in this broad body of biophysical research [9,22,27,28], this heterogeneity of adsorbed-water behaviour was manipulated also by externally applied electric fields interacting with implicit, intrinsic ones, the extent of dynamical crossover with HEWL per se examined (i.e., the age-old question of which dominates—protein or water), and, pertinently in the case of the present study, the hydration-layer IR characteristics.

One of the unanswered, open questions arising in the experimental, and, indeed, previous molecular-simulation, studies of hydrate proteins is the lack of careful, “onion-like” dissection of the hydration sub-layers, to scrutinise their individual characteristics and properties. This is perhaps understandable, as experimental humidity control is limited in its lower resolution to around 1–2 nm for hydration-layer thicknesses [15], which would typically yield about a half-dozen or more individual water sub-layers. Indeed, such “unpeeling” of the sub-layers, by analysing the underlying behaviours of each, allows one to ascertain the typical distance from the surface of the protein which results in the adoption of more bulk-like water properties. Indeed, given the “patchwork” surface characteristics of proteins, replete, inter alia, with distinct hydro-phobic and -philic patterns, the level of local electrostatic-potential is heterogeneous and non-uniform, which tends to lead to variable extents of (sub-)layer-structuring and character, and water molecules’ transitions from one to other.

Given that ubiquitin is such a well-studied and characterized protein [29], we felt that it would be appropriate to probe its dynamical characteristics and interactions with its surrounding hydration layer. Ubiquitin is centered about a core which is hydrophobic in character—made up of three residues in its α-helix and eleven residues from the β-sheet [29]. This core forms a strong template on which there is a large amount of hydrogen bonds—both internally to stabilize the protein (aside from the COOH terminus), and also with its surrounding hydration layer. Naturally, the role of water in determining the structural behavior of ubiquitin is important, in that the hydration layer helps to determine the local, underlying ubiquitin structure, as with other proteins [10,11].

With this in mind, we continue in the present work to study the dynamical properties of hydration water and the role of the hydrogen bonding at the interface in contact with proteins, using classical MD with a flexible water model. To address the gap in the literature vis-à-vis the characterisation of individual sub-layers, in the present study, we study ubiquitin, given that we have found that this displays rather well-defined water-density undulations and structure in its hydration layer—which is perhaps not surprising for its ready molecular surface recognition, given its ubiquitous role in regulating protein life-cycles. The water-density profile obtained from MD simulation, is related to the different solvation sub-shells of the overall “hydration-water layer”. This layer-unpicking behaviour can also be seen by studying, sub-layer by layer, features of the vibrational Density of States (VDOS) and infrared (IR) spectra; these have been studied for the first monolayer in immediate contact with these surfaces. The analysis of these spectra shows that the molecules of water in this first (sub-)layer adopt characteristics intermediate between movement-restricted/confined water and structural organisation characteristic of ambient-temperature liquid water. Due to the difficulties of realising a confined water simulation, which depends on the confinement environment, we use a comparison with ice as a gross approximation, only for reference purposes in terms of quantifying an extreme character of molecular confinement. In any event, as we shall soon see, these layer-by-layer vibrational and IR characterisations provide an intimate view of the protein/water interface, illustrating vividly the importance of the dynamical interplay between these sub-layers, in terms of enhancing our understanding of the fundamental mechanisms underpinning the protein-solvation process. To our knowledge, this is the first layer-by-layer study of this type, dissecting explicitly water VDOS and IR spectra in a representative protein’s close solvation environs for closely-adsorbed water molecules. As mentioned previously, advanced experimental-spectroscopy techniques typically have difficulty in probing within less than a nanometre of surfaces, which has arguably served to demotivate to date allied molecular simulation probing of individual sub-layers. 

## 2. Results and Discussion

### 2.1. The Protein-Hydration Water Shells: Radial Water-Distribution Density

As in our previous research, to identify the physically absorbed water molecules and the layer formed around them, the radial distribution of water density relative to the ubiquitin surface was calculated. Figure 1 shows four very different regions calculated using the atomically sensitive Voronoi-cell method, featuring two peaks near the protein surface. The centre of the first peak (d = 0.83 g/cm^−3^, distance from the protein = 1.77 Å) lies at the heart of the first region. As one would expect, it has the greatest interaction with the surface because is adsorbed directly, and forms strong hydrogen bonds with the ubiquitin, dominated by donors on the protein’s surface. The second, much more dense peak lies in the first transition zone (d = 1.12 g/cm^−3^, distance from the protein = 2.6 Å), hinging on the distinct density second peak and a long tail after the peak. In part, this more dense region features some weaker hydrogen bonds with the protein surface. An important characteristic of this higher packing density leads to “crowding” phenomena, where the second layer “pushes” or “presses” the first adsorbed layer towards the protein surface. Beyond these two peaks, we found two well-defined regions. The first one, a small area about 0.5 Å wide with a maximum density about 0.87 g cm^−3^, is called the third region. Further outward from the surface, a fourth zone appears (up to ~7.0 Å), which has a density slightly lower than that of the bulk water but serves as a starting point for subsequent adjustments. The third region is a transition zone, which acts as a ‘reservoir’ or ‘buffer zone’ between the internal ones. These zones have important interactions with directly adsorbed water molecules from the internal regions and the bulk water.

The radial water-density profiles obtained using the flexible Ferguson model evince a similar shape to our previous study on HEWL [28]. However, the two outer areas before the large bulk water region are more clearly distinguished. Due to the diffusion of water molecules, only the water molecules in each layer within at least 90% of each sampling interval can be used to extract the dynamical properties of the adsorbed-water layer.

### 2.2. Ubiquitin-Hydration Water Shells: Vibrational Density of States 

#### 2.2.1. Vibrational Density of States in Each Solvation Layer

The VDOS spectra calculated using Equation (2) for each of the solvation layers close to the surface are shown Figure 2 together with the spectrum of liquid water simulated using the Ferguson water model [28].

When comparing the VDOS obtained from liquid water, we found that the spectra obtained from different regions have similar shapes, but the difference is small, such as the position of the main peak.

In the protons’ libration region, the two characteristic bands of water appear, one at very low frequencies, around 50 cm^−1^, a second one from ~250 cm^−1^. The peak position in both bands is in a higher frequency position than that of water for the close solvation layers (first and second) moving towards a similar position in the other two layers. This blue-shift agrees with our previous theoretical study [28], for the second protons’ libration band, and also has been experimentally reported via THz spectroscopy [23,24], especially in the acoustic, optical and librational bands of solvated protein systems. In the H-O-H bending frequency range, we witness again this blue-shift for the closest-to-surface solvation layers. Finally, in the OH-stretching locale, the shape of the hydrogen-VDOS is quasi-equal to the liquid water. However, we determine that a shoulder appears in the first and second solvation layers around 3300 cm^−1^. This shoulder is clearer in the directly adsorbed water layer, although it is still somewhat accentuated in the second solvation layer where there is no direct contact with the protein. As we suggest in our previous study [28], this could be representative of a ‘OH-stretch-echo’ interplay in hydrogen bonds between the water molecules’ proton and primarily oxygen atoms on the protein surface that also affects the second layer, which confirms what we have seen previously—that the first and second layers are intimately influenced by the surface of the protein.

In general, for the case of oxygen atoms, we found a similar behaviour as described above for the hydrogen atoms. In the librational region, we found a substantial variation in the shape for the first solvation layer—a movement of 50 cm^−1^ for the high-intensity peak and the non-existence of the plateau so characteristic of liquid water and the rest of the layers. The librational blue-shift is similar to that described by Chakraborty et al. [30] for water molecules proximal to an α-helix. This finding is consistent with Raman and MD studies, which have indicated a small blue shift in the mid-IR OH band for water-solvating monosaccharides [31]. In the bending region, besides the blueshift, we also found that the two close surface layers exhibited a shoulder around 1700 cm^−1^. This shoulder could be related with water with movement limitations, as a similar one is evident for ice I_h_ spectra [30] in the same region for the same water model. Considering this, the closest-to-surface solvation spheres (first and second) are conditioned by interactions with the protein surface which induce a certain ordering, or enhanced structural order, of their constituent water molecules that restrict the water molecules’ internal movements.

Finally, in the ‘OH-stretch-region’ the profile resembles that of liquid water closely, except for the shoulder appearing for the two first layers at 3300 cm^−1^, as already described for the case of hydrogens. The fact that this shoulder also appears in the oxygen VDOS reaffirms that this interaction not only affects the molecules closest to the surface of the protein, but also affects the next solvation sphere. This type of behaviour, i.e., greater order and suppression of certain modes in water spectra, with regions qualitatively similar to those of ice, has been described previously for material/water interfaces, and is redolent of confined water in general [23,32,33]. The rest of the solvation layers exhibit intermediate behaviour to these, serving as a kind of buffer to allow water molecules to make transitions to and between individual areas and layers closer to the protein’s surface. This fact rationalizes the noticeably short periods of residence in these layers and that make these transitional buffer layers considerably narrower and dynamically varying in molecular make-up than their more intrinsically stable “internal-layer” counterparts.

#### 2.2.2. Vibrational Density of States: Summation of Layers 

The VDOS spectra was calculated at the protein/water interface as the summation of two, three and four layers, and are shown in comparison with the first layer and bulk water in Figure 3. The spectra that include more than one solvation layer were calculated by taking the mean from the summation of the water molecules’ spectra (for those that spend more than 90% of time within each layer, as mentioned previously). As in previously presented spectra herein, the three main regions are present, i.e., libration-rotational motions, H-O-H bending and O-H stretching. 

In general, it is clear that as the various solvation spheres are added, the spectrum obtained becomes closer and closer to that of liquid water. It can also be seen how the spectra of the first two layers continues to maintain almost all of the prominent characteristics already mentioned: blue-shift in all peaks, the shoulder in the ‘OH-stretch-region’. However, when the third solvation layer is added, some of these characteristics disappear, progressing the shape of the spectra to resemble liquid water when the last solvation layer is considered. This behaviour serves to confirm that the first and second solvation layers are influenced heavily by the protein, whilst the ones beyond serve as a buffer, a molecular transition zone, towards essentially bulk-like liquid water.

### 2.3. Ubiquitin-Hydration Water Shells: Infrared Spectra 

#### 2.3.1. Infrared Spectra in Each Layer

Using Equation (4), we calculate the IR spectra for each of the four solvation layers close to the external surface of the ubiquitin, which are shown in Figure 4. Similar to VDOS, we partitioned the IR spectra into three major absorption regions: a lower-frequency band for libration-rotational motions (under 1000 cm^−1^), the H-O-H bending vibration (near 1650 cm^−1^) and the O-H stretching vibration, which is one peak representing the combination of symmetric and asymmetric vibrational modes (between 3250 and 3750 cm^−1^).

We decomposed the low-frequency area in two parts—a first where peaks are absent from all spectral profiles at frequencies lower than 130 cm^−1^, and a second where the principal libration-rotational components appear for the presently used water model at frequencies up to ~1300 cm^−1^. It can be seen how the first three solvation layers are wider than the fourth (which has a width similar to that of liquid water). This may be related to the fact that these three layers adopt a hybrid shape between restricted-movement water (i.e., confined water or bonded water) and liquid water. This behaviour has been previously observed for the layer closest to the HEWL protein [28]. 

The shape for H-O-H bending is like that of liquid water; however, the first three layers of solvation display widths are greater than that of liquid water. In addition, as one moves away from the ubiquitin surface, it begins to be seen how the band unfolds into two components—one centered between 1600–1650 cm^−1^ and another between 1675–1700 cm^−1^, respectively. The second component may well be connected with restricted-movement water as the main component for ice I_h_ at bending vibration lies in this range [28,30,34]. 

Lastly, examining the band for O-H-stretching vibrations, we found similar behaviour to that in the previous, H-O-H bending one. Here, all bands are wider than those of liquid water. The same shoulder is also observed at 3300 cm^−1^ that was observed in the VDOS, which decreases as we move away from the protein. This shoulder does not appear in the interface IR spectra obtained for the HEWL [28].

#### 2.3.2. Summation of Layer-by-Layer Infrared Spectra 

The IR spectra calculated for the interface between water and the protein surface, in comparison with the summation of two, three and four layers, are shown in Figure 5. The spectra that include more than one solvation layer were calculated by the average-summation of the spectra of the water molecules that spend more than 90% of the time within each layer. As in previous spectra, the three main regions are present, i.e., libration-rotational motions, H-O-H bending and O-H stretching. 

In this case, the spectra obtained in this way have less fluctuations or noise than when we consider each of the solvation layers separately. In general, it is appreciated as the different solvation spheres are added, the spectra obtained approaches that of liquid water. In the low-frequency region, the same behaviour described above can be seen. In this case, when considering a greater region of solvation, different behaviour is not detected to any tangible extent. 

In the H-O-H bending-vibration region, the position of the bending-band centre displays a blue-shift as we move away from the protein. In the case of the first layer, this position is close to 1675 cm^−1^, whilst for the other three, it moves towards 1650 cm^−1^, This peak is related to the H-O-H bending vibration for liquid water. In addition, a shoulder appears at 1730 cm^−1^ in the first layer and starts to decrease as we add more solvation layers. This band and peak movement is related to a transition between low-mobility water molecules, via confinement or hydrogen bond interaction with the protein amino-acids towards a bulk-type liquid water [28,35]. For the last vibrational region considered, in the O-H-stretching band, the behaviour is quite like that of each of the separate layers. The 3300 cm^−1^ shoulder is located clearly, and there is a small decrease in the intensity, as we consider yet more solvation water molecules in this particular layer. 

Taken together, these results suggest that water molecules in the directly adsorbed layer do display a much more confined character than those in the other regions, between examination of the intermolecular and intramolecular vibrations. For intermolecular vibrations, e.g., acoustic, optic and librational regions, the spectra reflect suppression of the rotational and translational motion, together with common vibrational motions. In this respect, Persson et al. [36] have scrutinised how water molecules’ rotational movement are affected by the protein surface—with low and high and levels seen, reflecting the underlying hydro-phobic and –philic surface milieux. Low-confined water may rotate as a bulk-like collective, and highly confined water has longer residence times vis-à-vis unconfined water [28,36]. In any event, in the present work, we have found hints of this in the distinct spectra of the directly adsorbed water layer: the broader ‘shoulder’ in the 1st-layer O-VDOS spectra in Figure 2 and Figure 3 in the acoustic region (~100 cm^−1^) where we see this greater variety of water behaviour stemming from the blend of both the hydro-phobic and -philic surface locales compared to sub-layers further out from the surface which do not participate directly in protein-water hydrogen bonds.

## 3. Materials and Methods

Ubiquitin, 1UBQ [29], was extracted from the Protein Data Bank, and counter-ions were added to the water to render the whole system electroneutral. The protein was placed at the centre of a cubic periodic box with (x, y, z) dimension of approx. 60 Å, respectively, in the laboratory Cartesian frame of the original structure, with approx. 8000 molecules of water surrounding the protein structure. 

All MD simulations were performed using GROMACS version 5.1.2., University of Groningen, Groningen, The Netherlands [37] The approach used here is essentially an adaptation from that explained in our previous studies [9,22,27]; nonetheless, a brief account of the main differences is provided here for the sake of completeness and clarity. The OPLS force field [38,39] was used for the ubiquitin, whilst and the Ferguson flexible model [40] was adopted for water molecules, owing to its compatibility with the OPLS potential, together with its improved description of dynamic and electrostatic properties, without adversely influencing other properties. In addition, the functional form of the Ferguson model is reasonably like other well-validated and characterised SPC-type models. In addition, it should be noted that the use of flexible water models is important to examine the vibrational modes, such as VDOS and IR spectra—which, after all, is a fundamental goal of the present study, and explains further why we chose this flexible water potential model, as opposed to a rigid one, which would, *ipso facto*, not allow for the study of IR properties. However, in our choice of classical molecular dynamics as an overall approach, in conjunction with flexible water models, it is useful to consider this in the broader context of water/bio-surface interactions in an electronic-structure sense. In particular, the study of Ilawe et al. highlights the use of Density Functional Theory (DFT) to gauge peptide-water interactions, and glean useful insights into mixing of modes in hydration water and peptides [41]. The use of DFT requires smaller systems and, often, implicit solvent models, which often struggle to capture the shape of bio-surfaces and their key hydrogen-bond interactions owing to the lack of explicit water molecules to define these interactions. Still, DFT can, and often does, offer valuable insights into mode-coupling, although it is not feasible in the present study to model a system as large as ubiquitin with explicit-solvation resolution and sufficiently long simulations at appropriate temperatures for good statistical definition of vibrational modes, so the OPLS-Ferguson model is, on balance, the most appropriate method to use. 

The smooth particle-mesh Ewald (SPME) method was used to handle long-range electrostatic forces, as implemented in the GROMACS package. MD simulations were performed under three-dimensional periodic boundary conditions in the NVE ensemble via the Parrinello-Rahman approach in conjunction with velocity-Verlet integration with 0.4 fs time-steps (given that holonomic-constraint algorithm was applied for proton vibrations). The production simulations were performed for 10 ns, and the last 200 ps were analysed.

The density distribution of water molecules in the interfacial-layer environs of ubiquitin’s outer surface was calculated using state-of-the-art Voronoi cell analysis [42]. We computed the “Voronoi” volume associated with each molecule in the overall hydration layer, thus estimating the hydration-shell volume. We were then able to sub-divide the overall layer into distinct sub-layers based on the ‘kinks’ and extrema in its spatial density distribution (see below).

VDOS can be obtained through theoretical approaches [32,43], such as molecular dynamics. From MD simulation of water, the velocity autocorrelation function can be extracted, the Fourier transform of which yields the power spectra, or, equivalently, VDOS [43]. Moreover, through MD simulation, the IR spectra of water can be obtained via the dipole approach [44] or the electrical flux-flux autocorrelation function [44]. All else being equal, the introduction of bond flexibility in water models [19,40] generally improves the fidelity of calculation of the dielectric constant and leads to improved pressure-temperature-density behaviour, in closer accord with experiment [19].

The normalised velocity-ACF of an atom, *j*, (or particular group of atoms, α), $Z\left(t\right)$, is [34]:(1)$$Z^{\alpha }\left(t\right)=\langle v_{j}^{\alpha }\left(t\right)\cdot v_{j}^{\alpha }\left(0\right)\rangle /\langle v_{j}^{\alpha }\left(0\right)\cdot v_{j}^{\alpha }\left(0\right)\rangle$$
where brackets denote ensemble averaging and *i* denote the atom. The VDOS is the real part of the numerical Fourier transform of the (V)ACF; the underlying frequencies modes, *ω*, characteristic of time-variation may be gleaned therefrom [44]. The normalised VDOS (or power spectra) of the oxygen and hydrogen atoms in water were evaluated as:(2)$$Z\left(\omega \right)=\int_{0}^{\infty }Z^{\alpha }\left(t\right)\cos\left(\omega t\right)dt$$

The oxygen atoms in water dominate translational motion, whilst hydrogen atoms reflect librational motions [45]. VACFs, and the corresponding VDOS, were also tested against those computed from shorter NVE-sampling sub-trajectories, with little noticeable difference from their NVT counterparts. 

The infrared spectrum for the adsorbed water monolayer was obtained using the electrical flux-flux autocorrelation function [44] in a similar way to our previous study [32].
(3)$${\langle \sum }_{j=1}^{n}e_{j}v_{j}\left(t\right)\cdot \sum_{j=1}^{n}e_{j}v_{j}\left(0\right)\rangle =\langle \sum_{j=1}^{n}e_{j}\frac{dr_{j}\left(t\right)}{dt}\cdot \sum_{j=1}^{n}e_{j}\frac{dr_{j}\left(0\right)}{dt}\rangle =\langle \frac{dM\left(t\right)}{dt}\cdot \frac{dM\left(0\right)}{dt}\rangle$$
which corresponds to the ACF of the time derivative of the dipole moment [46]. The spectral density is then calculated as [34,38,47]:(4)$$I\left(\omega \right)\alpha \int_{0}^{\infty }\langle \frac{dM\left(t\right)}{dt}\cdot \frac{dM\left(0\right)}{dt}\rangle \cos\left(\omega t\right)dt=\int_{0}^{\infty }\langle \sum_{j=1}^{n}e_{j}v_{j}\left(t\right)\cdot \sum_{j=1}^{n}e_{j}v_{j}\left(0\right)\rangle \cos\left(\omega t\right)dt$$

To calculate the IR spectra of water near the water-protein interface only, the water molecules that spend 90% or more of the time in each sub-layer were identified, and these molecular sets taken into account for the definition of each (sub-)layers’ composition (often referred to hereinafter as “layer”, as opposed to the overall hydration layer). As mentioned previously, we needed to use a flexible water model to allow for determination of IR spectra, rather than rigid models.

In terms of data collection for subsequent analysis, time intervals of 20 ps were used for the first and second layer, and 5 ps were used for the rest, throughout the simulation, since there are no water-molecule chemisorption events at the protein, given the non-dissociative nature of the water potential used. After a good deal of testing, it was found important to use these interval durations to ensure that the same sub-population of molecules remained in each relevant layer for at least 90% of the interval duration—to maintain all-important consistency in “molecular membership” of each interval. All calculated dynamical properties correspond to the average of ten intervals, into which the last 200 ps of the trajectory was divided. All calculated spectra correspond to the average of spectra gathered over ten such intervals, and the duration of the simulations was checked in all cases to allow for such level of statistical data gathering.

In essence, the reason why different spectra were applied is that we wished these to act as diagnostic tools to characterize VDOS for different acoustic-like vibrational properties (as vibrational “fundamentals”), with these “pure” modes for O and H atoms in water being able to probe translational and librational, bend and stretch modes. IR serves as useful to compute, given that it has been measured experimentally for various proteins. 

## 4. Conclusions

MD simulations scrutinising the interaction between ubiquitin and its hydration-water sub-layers, by vibrational-mode and IR analysis have shown by VDOS that the first directly adsorbed sub-layer displays a confined character for water molecules therein. With increasing distance from the surface, layers adopt greater bulk-like spectral behaviour, indication that vibrational harmonisation to bulk occurs within 6–7 Å of the surface, at least from a vibrational standpoint.

It was found for ubiquitin that there is an important effect of hydration-layer water on vibrational properties, and that the confined water molecules in the hydration layers exhibit similar vibrational signature characteristic of water becoming more solid-like. In this sense, these ubiquitin results are more similar to our group’s previous finding for HEWL, with its patchwork-like surface of hydro-phobic and –philic regions [9,22,27,28], which is in contrast to more hydrophilic proteins, such as peripheral protein or glycophorin, where electrostatic interactions alone with the hydration layer tend to dominate.

The present scrutiny of the varied molecular ‘personality’ of the individual segments of the overall hydration layer emphasises the evident need to develop surface-sensitive experimental spectroscopy of high resolution with exquisite control of humidity to allow for ‘unpicking’ of the vibrational and IR phenomena in individual water layers, as well as the desirability for its further layer-by-layer MD characterisation. Indeed, we note the exciting prospect of both experiment and MD working hand in glove to build up, layer by layer, water shells atop each other, and study the quirks in the journey of adoption of more bulk-like properties—using a variety of metrics (beyond vibrational and IR). In this sense, the present study has taken the first tentative step in this layer-by-layer transition from tightly controlled “bio-water” to bulk water.

## Figures and Tables

**Figure 1 ijms-23-15949-f001:**
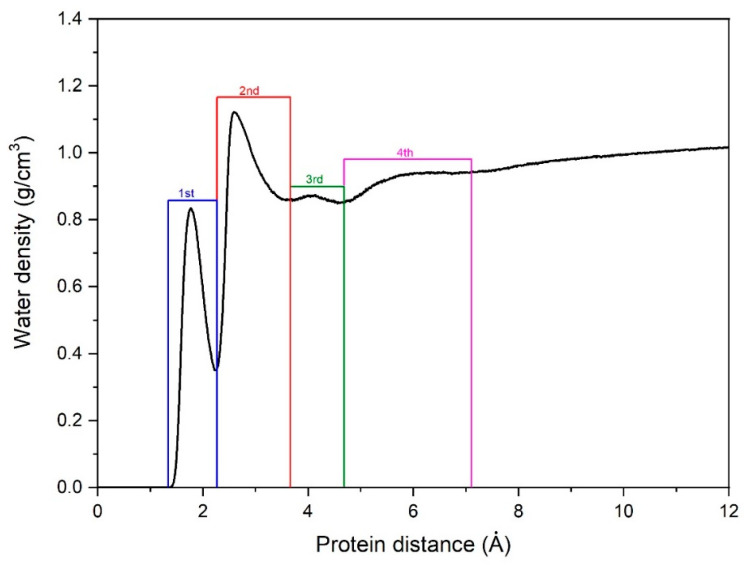
Radial density distribution of water molecules in the immediate environs of the ubiquitin surface. The four layers in closest proximity with the protein are marked in different colours.

**Figure 2 ijms-23-15949-f002:**
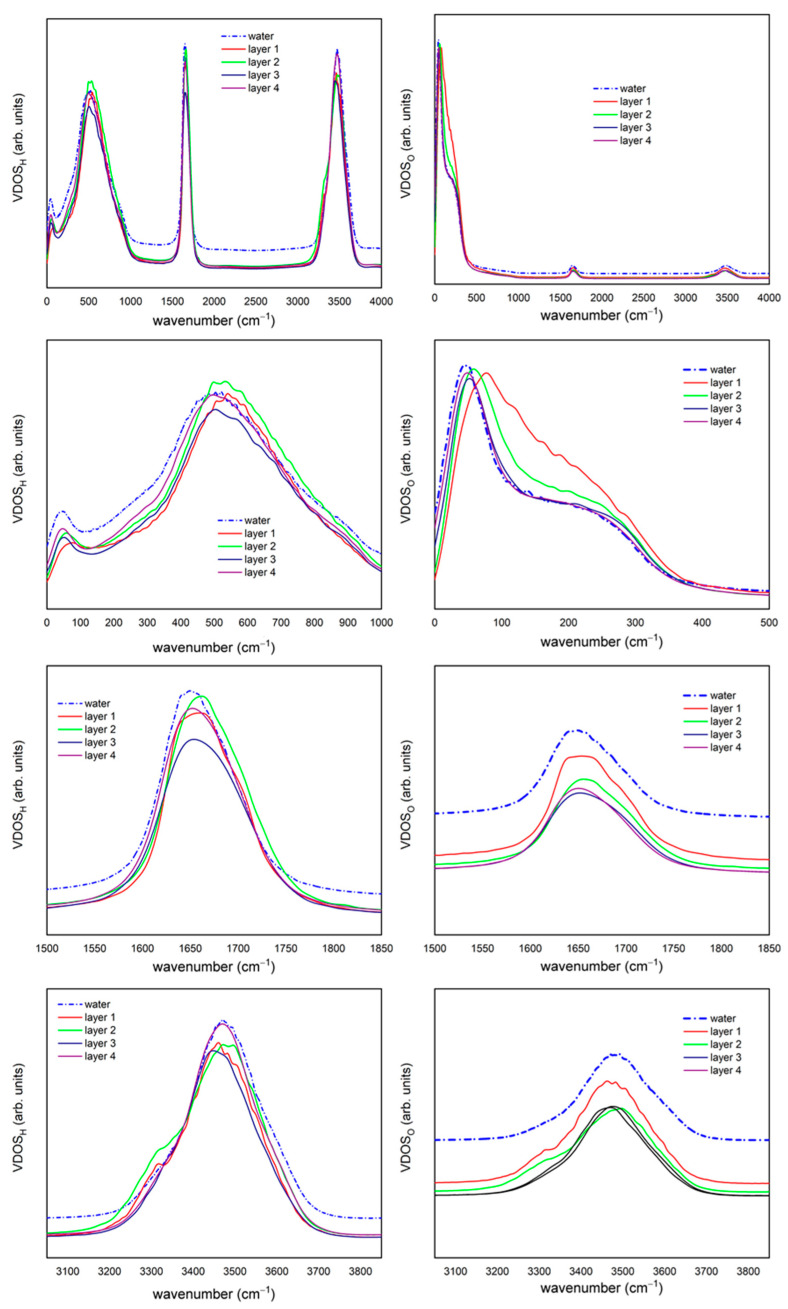
Vibration density of states for hydrogen (**left**) and oxygen (**right**) for first (red), second (green), third (dark blue), and fourth (purple) solvation sub-layers considered and liquid water (blue, line-dot-line). Full frequency range in the first row, the very low frequencies, e.g., acoustic, optic and librational regions, in the second row, H-O-H bending region in the third row and H-O stretching region in the fourth row.

**Figure 3 ijms-23-15949-f003:**
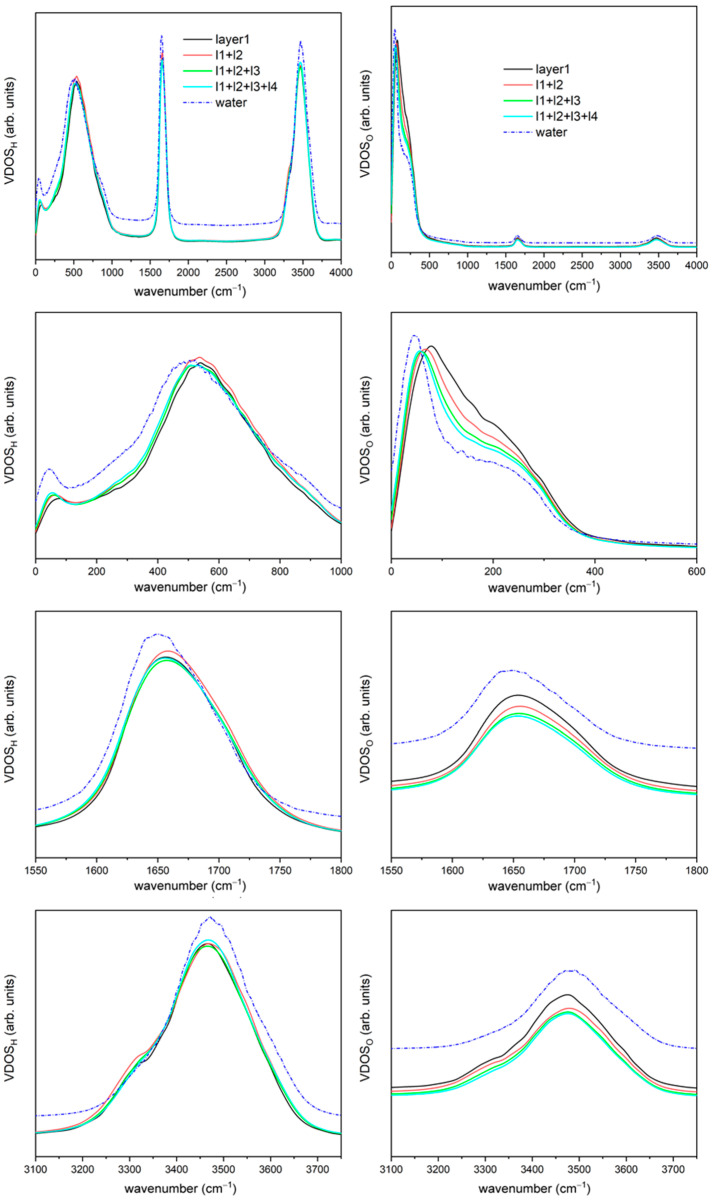
Vibration densities of states for hydrogen (**left**) and oxygen (**right**), for the water interfaces with one (black), two (red), three (green) and four (light blue) solvation layers in contact with the ubiquitin and liquid water (blue, line-dot-line). Full frequency range in the first row, the very low frequencies, e.g., acoustic, optic and librational regions, in the second row, H-O-H bending region in the third row and H-O stretching region in the fourth row.

**Figure 4 ijms-23-15949-f004:**
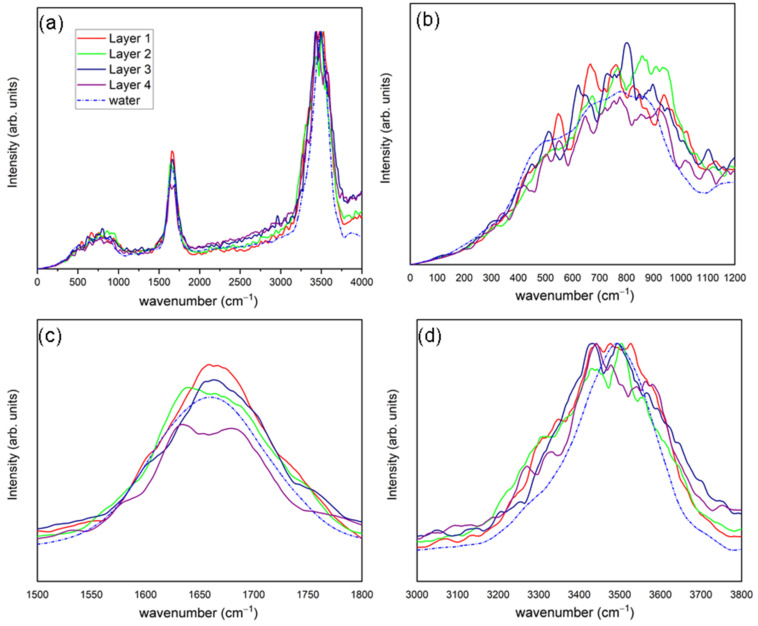
Full infrared spectra (**a**) and subsets for low-frequency region (**b**), H-O-H bending region (**c**) and H-O stretching region (**d**) for first (red), second (green), third (dark blue), and fourth (purple) solvation layers considered and liquid water (blue, line-dot-line).

**Figure 5 ijms-23-15949-f005:**
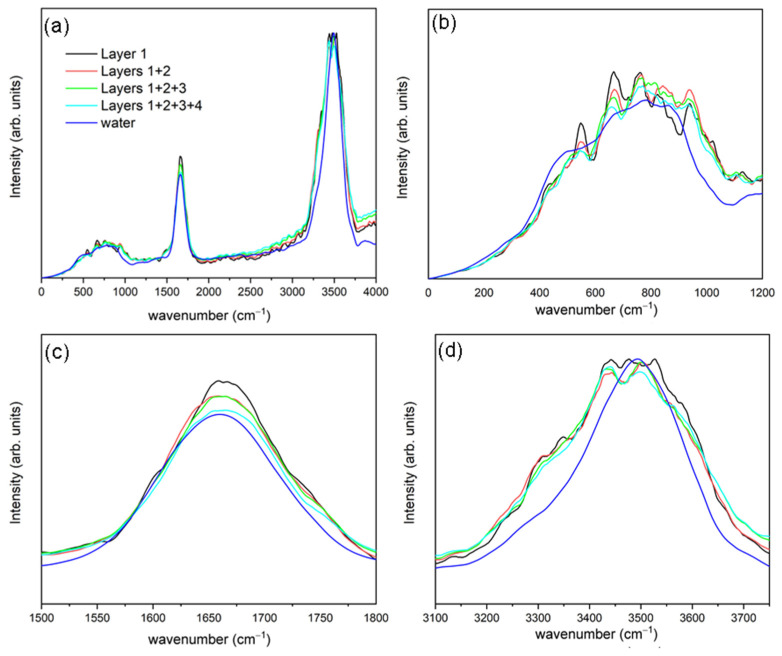
Full infrared spectra (**a**) and subsets for low-frequency region (**b**), H-O-H bending region (**c**) and H-O stretching region (**d**) for the water interfaces with one (black), two (red), three (green) and four (light blue) solvation layers in contact with the ubiquitin and liquid water (blue).

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
