# Peer review of "Vibrational Analysis of Hydration-Layer Water around Ubiquitin, Unpeeled Layer by Layer: Molecular-Dynamics Perceptions"

_ijms, 2022, doi:10.3390/ijms232415949_

Round 1

Reviewer 1 Report

The manuscript is devoted to the MD study of the ubiquitin dynamics in water solution, namely the structural organization of the  water shells close to the protein. There are several issues that should be clarified.

1. What is the reason of choice of Ferguson water model? What will change if parameters of water molecules changes to, for example, TIP3P, TIP4P etc. This is an important question as all conclusions are made from the MD simulations. Authors should provide results of additional calculations with other force field for water molecules.

2. It seems strange that the water density first peak is at 1.77A. Does this mean that there are only hydrogen bond donor groups on the protein surface and all water molecules are oriented to the protein surface by their oxygen atoms? Even in this case, the distance of 1.77A is too short. How can you explain this?

3. Terahertz experiments are definitely performed at constant temperature, but not constant energy of the system. How do you compare results obtained in the NVE ensemble with the experimental results. Solvent features depend on the temperature.

 4. What is the uncertainty / error of vibrational wavenumber estimates? Is this conclusion solid? “In addition, as one moves away from the ubiquitin surface, it begins to be seen how the band unfolds into two components - one centered between 1600-1650 cm-1 and another between 1675-1700 cm-1, respectively. The second component may well be connected with restricted-movement water as the main component for ice Ih at bending vibration lies in this range [28,42,43]”

5. Ref 3 -> decapitalize CHAPMAN

Reviewer 2 Report

Attached

Reviewer 3 Report

1) The text is really pleasant to read, however, its flowery language (rare latin expressions, elaborate words and sentences: "vide infra", "ceteris paribus", "lacuna", "rapprochement", etc.) can be a bit difficult or distracting for non-native readers. In my opinion, a scietific text should be easily available for a broad range of readers. However, it is not a significant remark and does not have to be addressed in the reviewed version of the manuscript with one exception: I do not consider "the lack of determination of solvation-layer size" an "existential problem" (page 2). The language here is too "flowery" this time. Please replace "existential" with "difficult" or any other similar word or już delete it.

2) IJMS is a general chemistry journal for a broad range of readers. It would be great if the authors could clearly explain somwhere in the "Methods" section the difference between the calculated VDOS and IR spectra in a few simple words and address following questions: 

- what is the reason for the differences in this type of spectra and what they mean?

- what is the reason for the separation of VDOS into O-VDOS and H-VDOS?

- how VDOS relate to the calculated FTIR spectra and possible experimental ones?

3) I am not convinced that the hydration water analysis in a layer-by-layer fashion is as novel as the authors state, however, it is still interesting to uncover properties of water in separate hydration layers of proteins. I see one problem, however, that is not clearly discussed in the text. To obtain a satisfactory spectra (VDOS or IR) the simulation time has to be long enough. Trajectories used in this paper are 20 or 5 ps long, not long enough. As I understand it, the spectra from several such simulations (200ps in total) were averaged. This action was probably aimed at ensuring that the water particles stay in a given layer for 90% of the time. This has to be clearly emphasized so that someone using such a protocol would not use this first value of 20ps (5ps).

Round 2

Reviewer 1 Report

I suppose that the text was not improved enough and questions from the first round remain the same.

Moreover, it is evident without any calculations  than water vibrational bands close to the protein surface should be different from the bulk as there are different hydrogen bond partners. It might be interesting to determine which particular interaction mostly affect vibrational frequency. But even in that case, it is evident that interactions with particles that have partial charges mostly differing from that of water molecules will contribute mostly.

Reviewer 2 Report

Attached

Round 3

Reviewer 1 Report

The Manuscript can be accepted in present form